# Neonatal Morbidities and Feeding Tolerance Outcomes in Very Preterm Infants, before and after Introduction of Probiotic Supplementation

**DOI:** 10.3390/nu14173646

**Published:** 2022-09-03

**Authors:** Ayoub Mitha, Sofia Söderquist Kruth, Sara Bjurman, Alexander Rakow, Stefan Johansson

**Affiliations:** 1Department of Medicine Solna, Clinical Epidemiology Division, Karolinska Institutet, 17177 Stockholm, Sweden; 2CHU Lille, Paediatric and Neonatal Intensive Care Transport Unit, Department of Emergency Medicine, SAMU 59, Lille University Hospital, F-59000 Lille, France; 3Women’s Health and Allied Health Professional Theme, Karolinska University Hospital, Solna, 17176 Stockholm, Sweden; 4Department of Women’s and Children’s Health, Karolinska Institutet, 17177 Stockholm, Sweden; 5Sachs Children and Youth Hospital, Södersjukhuset, 11883 Stockholm, Sweden; 6Department of Neonatology, Karolinska University Hospital, Solna, 17176 Stockholm, Sweden

**Keywords:** probiotic supplementation, very preterm infants, feeding tolerance, neonatal morbidities, necrotising enterocolitis, full enteral feeding, antibiotic

## Abstract

While probiotics are reported to reduce the risks of neonatal morbidities, less is known about probiotics and feeding tolerance. With this retrospective cohort study, we investigate whether introduction of probiotic supplementation as the standard of care was associated with fewer neonatal morbidities and improved feeding tolerance in very preterm infants. Using the Swedish Neonatal Quality Register, 345 live-born very preterm infants (28–31 weeks’ gestation), from January 2019–August 2021, in NICUs in Stockholm, Sweden, either received probiotic supplementation (*Bifidobacterium infantis*, *Bifidobacterium lactis*, *Streptococcus*
*thermophilus*) (139) or no supplementation (206); they were compared regarding a primary composite outcome of death, sepsis, and/or necrotising enterocolitis and secondary outcomes: time to full enteral feeding and antibiotics use. Probiotics seemed associated with a reduced risk of the composite outcome (4.3% versus 9.2%, *p* = 0.08). In the subgroup of 320 infants without the primary outcome, probiotics were associated with shorter time to full enteral feeding (6.6 days versus 7.2 days) and less use of antibiotics (5.2 days versus 6.1 days). Our findings suggest that probiotics improve feeding tolerance and further support that very preterm infants may benefit from probiotic supplementation.

## 1. Introduction

Many studies have shown that probiotics prevent diseases in preterm infants. While mechanisms of actions are not fully clarified, probiotics probably exerts their effect through reducing inflammation, improving the mucosal barrier, and promoting the microbiota development [1,2,3,4,5]. Less is known about whether probiotics improve feeding tolerance, a common problem in preterm infants [6,7,8]. Available data suggest that supplementation with probiotics improves feeding tolerance and shortens hospital stay [9,10,11]. A recent meta-analysis of randomized trials including almost 5000 preterm infants reported that probiotics reduced time to full enteral feeding by 1–2 days and reduced hospital stay by 3 days [12].

Given the large body of research showing that probiotics reduce the risk of necrotising enterocolitis (NEC) [13,14,15,16], routine use of probiotics is supported by independent societies, such as the Canadian Paediatric Society guidelines [17] and the European Society for Pediatric Gastroenterology, Hepatology and Nutrition (ESPGHAN) [18]. Based on the latter recommendations, the Swedish Neonatal Society issued a national guideline in 2020 based on the three probiotic strains supported by ESPGHAN [18,19]. Shortly thereafter, the four NICUs in the Stockholm region implemented probiotics as the standard of care for very preterm infants.

As a quality improvement project, we investigated the potential impacts for very preterm infants receiving probiotics as the standard of care, by comparing outcomes pre- and post-implementation, such as neonatal morbidities, need of an abdominal X-ray and antibiotics, feeding tolerance, and growth.

## 2. Materials and Methods

### 2.1. Population, Setting, and Data Source

This was a retrospective pre- and post-probiotics supplementation cohort study, based on all 369 very preterm infants (28 + 0–31 + 6 weeks’ gestation) live-born between January 2019 and August 2021, in the four NICUs in Stockholm County. Probiotics supplementation was introduced halfway through the study period. The study was based on data from the Swedish Neonatal Quality Register (SNQ), recorded at time of discharge by the discharging physician or through continuous recording by NICU staff during the length of stay. The completeness and validity of data in SNQ is excellent [20]. Medical records were used to validate recorded NEC diagnoses, to confirm whether probiotics supplementation was actually given to eligible infants and to extract secondary outcome data not available in SNQ. The study was approved by Swedish Ethical Review Authority (Dnr 2021-04296).

The study population included 345 infants, as we excluded 21 infants due to double entry in SNQ (*n* = 1), early transfer to a hospital outside Stockholm (*n* = 2), major non-NEC surgery (*n* = 1), or genetic disease or congenital malformation (*n* = 17), and 3 infants that died during the first week of life from causes related to pre- or perinatal events, such as birth asphyxia and IUGR, were excluded from the study population. Within the study population of 345 infants, there were none lost to follow-up until death (*n* = 2) or discharge (*n* = 343).

### 2.2. Enteral Nutrition Guidelines during the Study Period

The same enteral nutrition guideline was used in the Stockholm NICUs, and it underwent no revisions during the study period. Early feeding of very preterm infants was based on donated breast milk or mother’s own breast milk. Early formula feeding was only used in the rare instances when parents do not consent to donor milk. Delivered mothers were instructed to express colostrum and initiate pumping after delivery. Enteral feeding was to be initiated during the first hour after delivery, with the aim to reach 24 mL/kg/day the first day of life, and thereafter volumes were increased by 24 mL/kg or more per day. The guideline also stated that slower increments could be considered in cases of feeding intolerance. Full enteral feeding was generally based on mother’s own breast milk. Generally, some infants fully fed with donor milk, for example, due to low availability of mother’s own breast milk or due to maternal medication, may have been switched to preterm formula at postconceptional week 32–33 if bank milk supplies were low and prioritised for infants in need of early feeding with bank milk. Fortification of enteral feeds were tailored individually through a computer software application (Nutrium, Umea, Sweden).

### 2.3. Baseline Characteristics and Main Exposure

SNQ data was used to describe the study population. In April and May 2020, half-way through the study period, probiotic supplementation was introduced as the standard of care for very preterm infants in the four Stockholm NICUs. A freeze-dried blend of *Bifidobacterium infantis Bb-02* (DSM 33361), *Bifidobacterium lactis* (BB-12), and *Streptococcus thermophilus* (TH-4), a total of 1 billion CFUs in maltodextrin powder (ProPrems, Neobiomics, Solna, Sweden) was used, as suggested by ESPGHAN, diluted in 3 mL of breast milk or preterm formula. Per the guideline, probiotics were to be initiated during 2nd to 4th day of life, when 3 mL of enteral feeds were tolerated, and to be discontinued at post-conceptional week 34 + 0. The main exposure, probiotic supplementation, was also extracted from SNQ and validated in medical records. As expected, when implementing any new guideline, compliance to this probiotics guideline increased over time. Of eligible infants born in 2020 and 2021, 69% and 85% were supplemented, respectively. Overall, 139 of 180 eligible infants (77%) were supplemented with probiotics after the guideline was launched. As probiotics supplementation was our main exposure, we categorised the study population based on whether the infant had actually received probiotics or not.

### 2.4. Outcomes

Primary outcome was a composite of death, sepsis, and/or NEC. Deaths were recorded in SNQ, including cause and age at death, validated against the Swedish Death Registry. In order to consider a possible relationship between death and probiotics, death was defined as happening after the first week of life. Sepsis was defined as SNQ registration of at least one suspected/clinical or culture-proven sepsis. Suspected sepsis was defined as clinical instability, laboratory findings typical for sepsis, and treatment with antibiotics.

NEC was defined as NEC of Bell stage 2 or 3. For infants with a recorded diagnosis code (International Classification of Diseases 10th revision; CD-10) of P779 in SNQ, we validated and Bell-staged NEC diagnoses by reviewing information in the medical records (clinical signs, laboratory findings, abdominal X-ray results, need of surgery), unblinded to probiotics supplementation status.

Secondary outcomes were defined from SNQ data: weight change from birth to postnatal day 28 and post-conceptional week 36 + 0, days of parenteral nutrition, time to full enteral feeding ≥150 mL/kg/day, any antibiotics treatment, duration of antibiotics, and length of hospital stay. If the SNQ register had missing weights at 28 days and at 36 weeks, those were extracted from medical records, if available. All weights were standardised as Fenton z-scores [21]. Need and number of abdominal X-rays were extracted from medical records and defined as such, X-rays due to abdominal symptoms, when the referring clinician asked about radiological signs of NEC.

### 2.5. Statistical Analysis

Using Stata IC (Stata, v17, College Station, TX, USA), we calculated proportions and mean values for descriptive infant covariates. Associations between outcomes and probiotics use were tested with linear or logistic regression and were expressed as adjusted relative risk (aRR) with 95% confidence interval (CI). We adjusted for gestational age (days) and birth weight (Fenton z-score) as continuous values, based on the a priori assumption that these covariates were the most important confounders. Type of feeding was considered as a mediator rather than a confounder, on the causal pathway between exposure and outcomes, and was, therefore, not adjusted for. In sensitivity analyses, we calculated risks of the primary composite outcome including only surgical NEC cases (Bell stage 3). To investigate the impact of probiotic supplementation in the majority of more healthy preterm infants, we performed analyses for the secondary outcomes in the subgroup of infants without the primary composite outcome.

## 3. Results

Of 345 very preterm infants, 165 born before and and 180 born after probiotics supplementation were introduced to the standard of care. Compliance to the probiotics guideline, defined as percentage of infants getting probiotics after date of implementation in each hospital, was 77%. Therefore, and in total, 206 infants were not supplemented with probiotics, while 139 infants were supplemented with probiotics. Characteristics were similar between infants in both groups (Table 1). Comparing the 41 eligible but non-supplemented infants with the 139 supplemented infants showed no differences in baseline characteristics (Appendix A). Transfers between the Stockholm NICUs were very common. Among the 345 infants, 172 had recorded admissions to two NICUs, and 146 had recorded admissions to three NICUs. Most transfers were due to shortage of beds or due to neonatal homecare being provided by another NICU. No side effects were recorded from supplementation.

Twelve infants had an NEC ICD-code (P779) registered. Validation from medical records showed that four had NEC Bell stage 3, five had NEC Bell stage 2, and three were regarded as Bell stage 1 or misclassified. Consequently, the NEC cases in the no probiotics and probiotics groups were reduced from nine to seven and from three to two, respectively.

Two infants died, on day 8 and day 57, respectively, due to NEC, both occurring in the time period before probiotics were implemented as the standard of care.

The risk of the primary composite outcome of death, sepsis, or NEC, adjusted for gestational age and birth weight (Fenton z-score), seemed lower in the probiotics group (4.3% versus 9.2%, *p* = 0.08; aRR 0.44, 95% CI 0.18 to 1.08). While risk estimates suggested similar risk reductions for the separate outcomes in the composite, confidence intervals were wide due to few events per individual outcome (Table 2). We explored whether infant sex or mode of delivery contributed to the adjusted regression model. As neither was associated with the primary composite outcome (*p*-values 0.87 and 0.32, respectively) and point estimates did not change (data shown on request), we did not include those variables in the model. Furthermore, the risk reduction was practically unchanged in a sensitivity analysis only including surgical NEC cases (Bell stage 3) in the composite outcome (3.6% versus 7.8%, aRR 0.44, 95% CI 0.17 to 1.15; Appendix A).

Probiotic supplementation was associated with several secondary outcomes (Table 3). Need of parenteral nutrition and time to full enteral feeding were shorter, weight development from birth to postnatal day 28 and to week 36 were more favorable, need of an abdominal X-ray was less, and duration of antibiotics treatment was shorter. While differences were generally attenuated in the subgroup of 320 infants without the primary composite, probiotic supplementation was still associated with shorter time to full enteral feeding and shorter duration of antibiotics treatment (Table 4).

## 4. Discussion

Implementation of probiotics as the standard of care for very preterm infants (28–31 weeks) in Stockholm seemed associated with a reduced risk of the primary composite outcome of death, sepsis, and/or NEC. Furthermore, results suggested that implementation of probiotics improved feeding tolerance, as probiotics were associated with shorter time to full enteral feeding, more favorable growth, less need of an abdominal X-ray, and shorter duration for antibiotics treatment. Associations were attenuated in the subgroup of preterm infants without the primary composite outcome, but probiotics were still significantly associated with shorter time to full enteral feeding and shorter duration of antibiotics treatment.

Probiotic supplementation of preterm infants has been extensively studied. Our finding of a reduced risk of the composite outcome of death, sepsis, and/or NEC in this cohort of very preterm infants born in Stockholm aligns with the results from meta-analyses and systematic reviews of randomised trials and observational studies [13,14,22].

When looking into the separate outcomes of sepsis an NEC, we found that relative risks shifted in the same direction, but confidence intervals widened due to the small study population, with few events. Nevertheless, the point estimates within this cohort for sepsis and NEC suggested that use of probiotics were associated with risk reductions of a similar magnitude as found in a randomised placebo-controlled trial including 1099 infants born before 32 weeks [23]. This large trial also investigated a probiotic blend of 1 billion CFUs of *Bifidobacterium infantis Bb-02* (DSM 33361), *Bifidobacterium lactis* (BB-12), and *Streptococcus thermophilus* (TH-4) and found approximately 50% risk reductions, for NEC within the whole cohort and for late-onset sepsis within the subgroup of very preterm infants (28–31 weeks) [23]. Our findings that probiotics were associated with the reduced need of an abdominal X-ray by 41% and shorter antibiotics duration by 2 days support a positive association between probiotics and reduced morbidity risks. These results also deserve their own attention. Preterm infants may be more sensitive to the detrimental effect of ionising radiation because of the highly mitotic state of their cells [24]. To our knowledge, the association between probiotics and X-ray exposure is not previously reported. Furthermore, a shorter antibiotics duration probably reduces the impact of antibiotics on infant gut microbiota development, which may be beneficial to health in the short- and long-term [25]. In fact, less use of antibiotics should be a goal in every NICU and, if verified in other studies, one may consider probiotic supplementation as a component of good antibiotics stewardship [26].

Probiotic supplementation was also associated with shorter duration of parenteral nutrition, shorter time to reach full enteral feeding, and more favorable weight development. These findings were similar with widened confidence intervals among preterm infants without the primary composite outcome, suggesting that probiotic supplementation also improved feeding tolerance in healthier very preterm infants, a subgroup that constitutes the majority of infants born very preterm. Previous research has shown that probiotics reduce the incidence of feeding intolerance and shorten the time to reach full enteral nutrition [27,28]. In a network meta-analysis that included 45 randomised controlled trials (RCTs), comparing the efficacy of different probiotic supplements in preterm infants, bifidobacterium plus lactobacillus led to a reduction in time to full enteral feeding [28].

The strength of our study is that the study population was a recent cohort of preterm infants born in the largest Swedish region, with only four NICUs sharing clinical guidelines. The study was based on information recorded prospectively through a standardised data collection for the Swedish neonatal quality register (SNQ). All infants with a recorded ICD-10 code for NEC went through a validation process based on clinical, laboratory, and radiology findings in the medical records. Secondary outcomes were investigated in a sensitivity analysis restricted to the subgroup of infants without the primary composite outcome.

We also acknowledge several limitations. While regional guidelines related to enteral nutrition did not change during the study period, the nature of an observational and unblinded design may have induced information bias. The small study population and lack of maternal data, for example, on chorioamnionitis and antenatal steroids, restricted the possibilities for more advanced analyses. Therefore, the reported associations may be subjected to residual confounding. Our a priori assumption was that the most important confounders were gestational age and birth weight for gestational age, but we explored adjustments with infant sex and mode of delivery and found neither to be contributing to the model or changing the point estimates. When it comes to mode of delivery, it is also reported that preterm birth and etiology of preterm birth, rather than mode of delivery, have an impact on the microbiota development in preterm infants [29,30]. One also needs to interpret composite outcomes with caution, especially if the individual outcomes are not competing or drive associations in opposite directions [31,32]. However, it could be hard to distinguish NEC from sepsis, and both those morbidities are also common causes of death in preterm infants [33]. In our dataset from SNQ, we did not have information about dates for NEC and/or sepsis. Another limitation is that misclassification of NEC diagnoses is not uncommon in clinical practice [34]. Misclassification of outcome cannot be excluded in this study. While following the diagnostic Bell criteria, our validation of NEC diagnoses through medical record data was unblinded. However, we performed sensitivity analyses of a composite outcome only including surgical NEC cases and found the associations to be very similar.

## 5. Conclusions

Implementation of probiotics as the standard of care for very preterm infants (28 + 0–31 + 6 weeks) in Stockholm seemed to be associated with a reduced risk of neonatal morbidities and was also reflected in less need of an abdominal X-ray and shorter duration for antibiotics treatment. Furthermore, probiotics were associated with improved feeding tolerance, and more favourable growth. Given the small study population and the observational and unblinded design, our findings need to be interpreted with caution, until replicated in a larger study. Moreover, those associations should not be generalised to extremely preterm infants. However, a logical next step would be to investigate whether probiotics impact such “softer” outcomes within a more vulnerable and immature population of preterm infants.

## Figures and Tables

**Table 1 nutrients-14-03646-t001:** Characteristics of 345 very preterm infants born in Stockholm before and after the implementation of routine probiotics supplementation.

Perinatal Characteristics ^a^	No Probiotics (*n* = 206)	Probiotics (*n* = 139)	*p*-Value
Gestational age, weeks (SD)	30.3 (1.1)	30.2 (1.1)	0.50
Birth weight, Fenton z-score (SD)	0.0 (0.9)	0.02 (0.8)	0.78
Birth weight for gestational age, *n* (%) - Appropriate - Large - Small	176 (85.4) 9 (4.4) 21 (10.2)	122 (87.8) 5 (3.6) 12 (8.6)	0.82
Male, *n* (%)	113 (54.8)	68 (48.9)	0.28
Singleton, *n* (%)	155 (75.2)	100 (71.9)	0.49
Vaginal delivery, *n* (%)	45 (21.8)	41 (29.5)	0.11
Apgar score < 7 at 5 min ^b^, *n* (%)	35 (17.2)	18 (13.0)	0.29
Transient tachypnea, *n* (%)	89 (43.2)	60 (43.2)	0.99
Respiratory distress syndrome, *n* (%)	86 (41.8)	56 (40.3)	0.79
Invasive ventilation ^c^, days (SD)	4.5 (6.4)	2.4 (2.6)	0.18
CPAP/HFNC ^d^, days (SD)	13.2 (15.2)	13.4 (14.6)	0.90
Bronchopulmonary dysplasia, *n* (%)	10 (4.8)	4 (2.9)	0.36
Patent ductus arteriosus, *n* (%)	2 (1.0)	4 (2.9)	0.18

CPAP (continuous positive airway pressure), HFNC (high flow nasal canula), ^a^ presented as mean value and standard deviation, or number of infants and proportion, ^b^ missing data *n* = 4, ^c^ based on 33 and 19 infants needing any invasive ventilation in the no probiotics and probiotics groups, respectively, ^d^ based on 202 and 128 infants needing CPAP and/or HFNC in the no probiotics and probiotics groups, respectively.

**Table 2 nutrients-14-03646-t002:** Rates and risks of the composite outcome of death, sepsis, and/or necrotising enterocolitis, in 345 very preterm infants.

	No Probiotics *N* = 206 *n* (%)	Probiotics *N* = 139 *n* (%)	Crude RR (95% CI)	Adjusted RR ^a^ (95% CI)
Composite outcome of death, sepsis and/or necrotising enterocolitis	19 (9.2)	6 (4.3)	0.47 (0.19–1.14)	0.44 (0.18–1.08)
Single outcomes of death, sepsis, or necrotising enterocolitis				
- Death	2 (1.0)	0 (0)	-	-
- Sepsis	15 (7.2)	5 (3.6)	0.49 (0.18–1.33)	0.46 (0.17–1.23)
- Necrotising enterocolitis	7 (3.4)	2 (1.4)	0.42 (0.09–2.00)	0.41 (0.08–1.96)

^a^ risks adjusted for gestational age (days) and birth weight (Fenton z-score).

**Table 3 nutrients-14-03646-t003:** Feeding tolerance, growth, and other secondary outcomes, in 345 very preterm infants.

	No Probiotics (*n* = 206)	Probiotics (*n* = 139)	Crude RR (95% CI)	Adjusted RR ^a^ (95% CI)
Parental nutrition ^b^, days (SD)	8.2 (7.7)	7.0 (3.8)	−1.22 (−2.65–0.20)	−1.43 (−2.76–−0.10)
Time to enteral feeding ≥150 mL/kg/d ^c^, days (SD)	7.4 (3.2)	6.8 (3.3)	−0.66 (−1.37–0.04)	−0.73 (−1.39–−0.07)
Weight change from birth to day 28 ^d^, Fenton z-score (SD)	−0.71 (0.41)	−0.61 (0.41)	0.10 (0.02–0.19)	0.11 (0.02–0.19)
Weight change from birth to 36 weeks ^e^, Fenton z-score (SD)	−0.72 (0.49)	−0.60 (0.49)	0.11 (0.01–0.22)	0.12 (0.02–0.22)
Length of stay in the NICU, days (SD)	39.6 (17.1)	38.6 (13.8)	−0.98 (−4.41–2.44)	−1.67 (−3.97–0.63)
Length of stay NICU and homecare, days (SD)	60.3 (21.0)	59.1 (15.4)	−1.16 (−5.26–2.94)	−1.82 (−4.96–1.31)
Abdominal X-ray, *n* (%)	58 (28.2)	24 (17.3)	0.61 (0.40–0.94)	0.60 (0.39–0.90)
Antibiotics treatment, *n* (%)	131 (63.6)	80 (57.6)	0.90 (0.76–1.08)	0.88 (0.76–1.04)
Duration of antibiotics ^f^, days (SD)	7.6 (6.8)	5.7 (2.8)	−1.82 (−3.39–−0.25)	−2.06 (−3.63–−0.50)

^a^ risks adjusted for gestational age (days) and birth weight (Fenton z-score), ^b^ missing data *n* = 17, ^c^ missing data *n* = 5, ^d^ missing data *n* = 3, ^e^ missing data *n* = 4, ^f^ based on 131 and 80 infants given antibiotics in the no probiotics and probiotics groups, respectively.

**Table 4 nutrients-14-03646-t004:** Feeding tolerance, growth, and other secondary outcomes in 320 very preterm infants without death, sepsis, and/or necrotising enterocolitis *.

	No Probiotics (*n* = 187)	Probiotics (*n* = 133)	Crude RR (95% CI)	Adjusted RR ^a^ (95% CI)
Parental nutrition ^b^, days (SD)	7.0 (5.2)	6.6 (3.1)	−0.36 (−1.39–0.66)	−0.61 (−1.52–0.30)
Time to enteral feeding ≥150 mL/kg/d ^c^, days (SD)	7.2 (2.9)	6.6 (2.9)	−0.60 (−1.25–0.05)	−0.69 (−1.29–−0.09)
Weight change from birth to day 28 ^c^, Fenton z-score (SD)	−0.69 (0.39)	−0.61 (0.41)	0.07 (−0.01–0.17)	0.08 (−0.01–0.16)
Weight change from birth to 36 weeks ^c^, Fenton z-score (SD)	−0.69 (0.49)	−0.60 (0.49)	0.09 (−0.02–0.20)	0.08 (−0.02–0.19)
Length of stay in the NICU, days (SD)	38.2 (15.3)	38.1 (13.8)	−0.12 (−3.40–3.16)	−1.12 (−3.27–1.04)
Length of stay NICU and homecare, days (SD)	59.7 (20.0)	58.8 (15.5)	−0.92 (−5.00–3.15)	−1.96 (−5.01–1.08)
Abdominal X-ray, *n* (%)	41 (21.9)	21 (15.8)	0.72 (0.45–1.16)	0.69 (0.43–1.10)
Antibiotics treatment, *n* (%)	113 (60.4)	74 (55.6)	0.92 (0.76–1.11)	0.89 (0.75–1.06)
Duration of antibiotics ^d^, days (SD)	6.1 (4.0)	5.2 (2.1)	−0.83 (−1.83–0.18)	−1.11 (−2.10–−0.12)

* excludes infants with the composite outcome death, sepsis, and/or necrotising enterocolitis (*n* = 25), ^a^ risks adjusted for gestational age (days) and birth weight (Fenton z-score), ^b^ missing = 17, ^c^ missing = 2, ^d^ based on 133 and 74 infants given antibiotics in the no probiotics and probiotics group, respectively.

## Data Availability

The dataset cannot be shared due to the regulations around Swedish national quality registers. STATA code can be provided upon request.

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
