# Peer review of "Neonatal Morbidities and Feeding Tolerance Outcomes in Very Preterm Infants, before and after Introduction of Probiotic Supplementation"

_nutrients, 2022, doi:10.3390/nu14173646_

Round 1

Reviewer 1 Report

I read with interest the manuscript intitled “Neonatal morbidities and feeding tolerance outcomes in very 2 preterm infants, before and after introduction of probiotic supplementation” by  Ayoub Mitha & al. This study is well designed and provide some new evidence on the interest of probiotics in VLBW infants.

However, the present study had some limitations:

1- It is not a DBRCT but a retrospective cohort study evaluating pre/post- probiotic supplementation.

2- Primary outcome was a composite of Death, sepsis, /and or NEC. However, all deaths occurred in the first period before probiotic and 4/5 during the first 2 wks of life. Per the guideline, probiotics were introduce during the 2nd to 4th days of life. According to results, at least 2 of the died were the result of antenatal or prenatal events that cannot be prevent by the introduction of probiotics 2 to 4 DOL (birth asphyxia or IUGR/renal failure). These 2 patients would be excluded from the analysis. For the 3 other patients with volvulus or NEC post-natal age of the event needs to be specified. Similarly, date of event and additional information should be provide for NEC and sepsis in the two groups to confirm there possible relationship to the probiotics treatment period.

3- It is generally admitted that own mother milk administration reduces NEC and Sepsis incidence in VLBW infants compared to donor milk, cow milk based HMF or PTF. According to protocol, early feeding of very preterm infants was based on donated breast milk or mother’s own breast milk. Early formula feeding was only used in the rare instances when parents do not consent to donor milk. Fortification of enteral feeds were tailored individually. Therefore, the risk of primary composite outcome should be also adjusted to the type of feeding during the period of the primary outcome incidence.

4- Secondary outcomes were: weight change from birth to post-natal day 28 and post-conceptional week 36+0, days of parenteral nutrition, time to full enteral feeding ≥150 ml/kg/d, any antibiotics treatment, duration of antibiotics, breast milk feeding at discharge, and length of hospital stay,….. Several of these secondary outcomes could be directly influenced by the feeding regimen during the study; own mother milk, donor milk, cow milk based HMF or PTF. By contrast, breast milk feeding at discharge could be poorly related to the prebiotics treatment but due to low availability of mother’s own breast milk or due to maternal medication, interruption of donor milk support…We suggest that incidence of breast milk feeding at discharge were exclude from the secondary outcomes but that the secondary outcomes were evaluated adjusted for the type of feeding ; exclusive own mother milk, percentage of donor milk or percentage of HMF or PTF.

Reviewer 2 Report

Why  are there no p values for the regressions?

Table 4 is redundant:

Breast milk feeding at discharged, n (%) No 61 (30.1) 28 (20.0) Reference Reference Yes, partial 132 (65.0) 97 (69.3) 1.06 [0.91 – 1.23] 1.06 [0.92 – 1.23] Yes, exclusive 10 (4.9) 15 (10.7) 2.19 [1.01 – 4.73] 2.20 [1.02 – 4.76] Breast milk feeding at discharged, n (%) No 61 (30.1) 28 (20.0) Reference Reference Yes, partial or exclusive 142 (70.0) 111 (79.9) 1.14 [1.01 – 1.29] 1.14 [1.01 – 1.29]

Reviewer 3 Report

Interesting quasi-experimental study evaluating the role of probiotic supplementation on feeding tolerance and other clinical outcome in a cohort of Swedish prematures. 

Some minor commnets to the authors:

1. Please, make some comments on the reasons tho choose that blend of probiotics.

2. Could you make any comparison regarfding the type of feeding (mother's milk, donor milk, formula?

3. As a footnote the significance  of the abbreviations in the table should be provided

Round 2

Reviewer 1 Report

First, we want to thank the authors for their answers to our comments and the revision of the manuscript. Nevertheless, we have still some comments and are not convince that the present study had the power to evaluate the potential benefit of probiotic supplementation on their primary and secondary objectives.

Per the guideline, probiotics were to be initiated during, when 3 ml of enteral feeds were tolerated, and to be discontinued at post-conceptional week 34+0. Compliance to this probiotic’s guideline was 77%, i.e. 139 of 181 eligible infants was supplemented with probiotics after the guideline was launched. 42 infants of the probiotic period were considered non-compliant to the probiotic guideline and included in the non-supplemented group (164+42=206).

However, non-compliant don’t mean that they don’t receive any supplementation during the interval between 2nd to 4th day of life and at post-conceptional week 34+0.  How many of those 42 infants received some probiotic supplementation?

Thus, by protocol, all preterm infants selected in the supplemented group needed to be alive at 24+0 PCA by contrast to the preterm included in the non-supplemented group. This methodology induces a bias in favor of the supplemented group. So, the 2 populations differ significantly and some of the added 42 infants could be partly supplemented with probiotic!!! Data on the non-compliant group needs to be provided

Therefore, It could be preferable to use an intend to treat statistical analysis to evaluate if the introduction of the probiotic supplementation induces a beneficial effect on dead and morbidities independently of the protective effect of HM.

According to protocol, type of feeding was only registered at discharge, and outcomes occurred before discharge. Unfortunately, this longitudinal information was not extract from the medical records and cannot be integrate in the analysis. Thus, it is not possible to evaluate the relationship between feeding regimen versus outcome.

 Nevertheless, I do not see a valid reason to include exclusive HM in the secondary outcomes. The use of breastfeeding at discharge is multi factorial as suggested in the manuscript (mother treatment, insufficiency of HM, social condition, clinical status of the preterm in the NICU….), all factors that are not related to the probiotic supplementation. By contrast, all the secondary outcomes could be directly positively influenced using exclusive or partial HM during the study period. Thus, exclusive HM at discharge cannot be considered as a secondary outcome but the HM volume or proportion received during the study is an unmeasured confounder biasing the interpretation of the probiotic effect

As a conclusion we suggest that the two populations are not similarly selected using exclusive compliant preterm infants at 34 wks GA as the inclusion criterion in the supplemented group by contrast to the non-supplemented group (as shown by the inclusion of 4 preterm infants dead before 10 days of life in the original manuscript). In addition, the inclusion of the 42 non-compliant preterm infants in the non-supplemented group induces an additional bias in this study. Reasons for non-compliance could be directly related to one of the primary or secondary outcomes (septicemia, need for PN, gastro-intestinal disorders…) and are not provided in the manuscript.

Due to the use of the data from the Swedish Neonatal Quality Register (SNQ) it was not possible to record the precise feeding regimen during all the study period and to evaluate the respective effects of HM and probiotics on the primary and secondary outcomes in preterm infants.
